# A Comparative Research on Corrosion Behavior of Electroplated and Magnetron Sputtered Chromium Coatings

Stefano Mauro Martinuzzi [1,2,3], Lorenzo Donati [1], Walter Giurlani [1,2], Federico Pizzetti [1], Emanuele Galvanetto [4], Nicola Calisi [4], Massimo Innocenti [1,2,3,5] and Stefano Caporali [2,4,*]

1 Department of Chemistry "Ugo Schiff", University of Florence, 50019 Sesto Fiorentino, Italy; ste.martinuzzi@gmail.com (S.M.M.); lorenzo.donati4@stud.unifi.it (L.D.); walter.giurlani@unifi.it (W.G.); federico.pizzetti.97@gmail.com (F.P.); m.innocenti@unifi.it (M.I.)
2 National Interuniversity Consortium of Materials Science and Technology (INSTM), 50121 Firenze, Italy
3 Center for Colloid and Surface Science (CSGI), 50019 Florence, Italy
4 Department of Industrial Engineering (DIEF), University of Florence, 50139 Firenze, Italy; emanuele.galvanetto@unifi.it (E.G.); nicola.calisi@unifi.it (N.C.)
5 Institute of Chemistry of Organometallic Compounds (ICCOM)—National Research Council (CNR), 50019 Sesto Fiorentino, Italy
* Correspondence: stefano.caporali@unifi.it

**Abstract:** Chromium deposits rank among the most widespread metal coatings for functional and decorative purposes. However, electrodeposition from Cr(VI) aqueous solution, which is the industrial process currently used to achieve thin chromium deposits, presents several environmental issues, urging the development of alternative solutions. In this study, which aims to compare the functional and aesthetical properties of decorative coatings, 800 nm thick chromium layers were deposited directly onto copper substrates by means of magnetron sputtering (MS) and direct current electrodeposition (ED). The obtained coatings were characterized regarding their thickness, morphology, color index, chemical composition, and corrosion resistance viewpoints. With respect to the electroplated samples, which constitute the benchmark, the magnetron sputtered ones appeared to be darker but showed a pleasing reflective appearance and superior thickness homogeneity, as well as the absence of voids and macrocracks, as evidenced by scanning electron microscopy (SEM) investigation. Regarding the anticorrosion performances of the chromium layers, electrochemical and standard salt spray test (NSS test) were carried out at room temperature in 3.5% NaCl solution, and evidenced similar, if not better, results for the magnetron sputtered coatings. On the whole, the data displayed here demonstrates the suitability of magnetron sputtering technology for use as an industrially feasible finishing technology for both aesthetic and mild-condition anticorrosion purposes.

**Keywords:** magnetron sputtering; chromium; SEM; coatings; corrosion; thin film

## 1. Introduction

A pleasing reflective appearance together with significant corrosion resistance, hardness, low friction, and durability, are among the typical properties of metallic chromium. As a result, the deposition of thin, usually below 1 μm, chromium layers is widely used as a finishing treatment to improve the functional and aesthetical quality of everyday life articles as well as some industrial items [1–3]. The most common commercial type of decorative chromium coatings range from 0.25 to 1.0 μm [3,4] and are generally achieved via electrodeposition from Cr(VI) aqueous solutions. Typically, these coatings are realized onto other electrodeposits, such as nickel, but are occasionally plated directly over the substrate of the part [5,6]. Unfortunately, while metallic and trivalent chromium are relatively benign species, hexavalent chromium is a well-known human carcinogen [7] in addition to creating other health concerns such as deviated septa and skin and lung irritation [8]. This is the

reason for the ban on processes based on hexavalent chromium in the European Union and the USA, urging worldwide companies to look for less hazardous alternatives [9–11].

All things considered, there are two different technologically feasible approaches to fulfilling this obligation: (1) substituting the hexavalent chromium with other, less toxic, chromium species such as Cr(III), and (2) depositing chromium directly from a metallic source by means of gas-phase techniques, thus avoiding, in such a way, the use of chromium ions at all.

The first approach is the most straightforward solution. Indeed, electroplating from Cr(III) aqua complexes such as $[Cr(H_2O)_6]^{3+}$ have been used for plating chromium layers; however, this process requires high temperatures and high voltages [12], resulting in a slow and highly inefficient process due to intense hydrogen evolution, leading to the formation of barely suitable coatings in terms of aesthetics and functionality [13–16]. Due to the presence of organic complexing agents in the plating baths, Cr coatings obtained from Cr(III) solutions are characterized by the presence of carbides formed during the electrodeposition process, which are responsible for the formation of microcracks that drastically decrease the coatings' anticorrosion properties [17]. Additionally, the aesthetical properties of coatings derived from Cr(III) baths are reduced by the presence of the abovementioned carbides [18] as well as some inorganic contaminants such as nickel and iron [6], which also negatively affect their anticorrosion properties [19].

On the other hand, chromium electrodeposition from ionic liquids (ILs) and deep eutectic solvents (DES) has also been reported [20–22]. It was thought that by using these non-aqueous media some of the abovementioned obstacles would be overcome, but issues regarding the industrial application of these technologies, which require dedicated plants and careful wastes management, remain. For instance, as a direct consequence of the high viscosity of these electrochemical baths, both solution losses and potential severe environmental issues due to higher drag-out must be considered.

Regarding non-electrochemical methods used to deposit chromium coatings, thermal spray HVOF (High Velocity Oxy Fuel), 3D laser-melt coating, and atmospheric plasma spaying could be used for industrial applications, especially to substitute hard and thick (up to 100 µm) chromium plating, such as for the protection of nuclear fuel elements [23]. However, these cannot be considered as viable alternatives for decorative purposes due to the relative high roughness of the obtained deposits.

Vice-versa, techniques based upon physical vapor deposition (PVD) allow for the deposition of a thin layer with smooth surfaces, presenting good mechanical and tribological properties. For such a reason, several PVD-based techniques are currently used, with a wide range of already established applications [24], especially for aesthetical purposes as a finishing treatment [25,26]. In PVD processes, the material to be deposited is transformed into atomic particles by a thermal physical process of collision and directed to the substrates where they condense to form a solid coating [27]. The process is typically carried out in low-pressure and low-temperature chamber, avoiding the internal stresses typical of electrodeposited [28] and high temperature sputtered coatings [29].

While many types of sputtering processes are available, the most common is magnetron sputtering (MS). This technique uses the confinement of the exiting plasma by a strong magnetic field that results in high deposition rates and superior adhesion properties compared to vacuum evaporated films [30,31]. On the other hand, MS coatings also present some drawbacks, such as the formation of defects due to the columnar growth mechanism, which may cause the formation of micropores and pinholes that would negatively affect the tribological and corrosion resistance of the coatings [30,31].

Indeed, the use of MS for the deposition of Cr-based coatings, such as Cr-alloys and CrN, for industrial and technological applications, has been already reported [32,33]. Nevertheless, comparative studies on the morphological and structural characteristics of layers derived from MS and electrochemical deposition methods are rather scarce in the literature [20–22]. Although the pros and cons of the two different deposition methods are well known, specific considerations may apply to the particular system under

investigation, impairing the straightforward generalization of their characteristics such as color, roughness, type of defects, and corrosion behavior [34,35].

In this context, the purpose of this work was to examine the characteristics and physicochemical properties of industrially feasible pure chromium coatings produced by both electrodeposition (ED) and magnetron sputtering (MS). In doing this, thin films of the same nominal thickness (0.8 μm) were deposited onto the same copper substrates and the characteristics of these coatings, such as thickness homogeneity, color, and anticorrosion properties were evaluated and rationally compared in order to assess the suitability of MS-Cr coating as an industrially feasible alternative to Cr(VI) electroplating.

## 2. Materials and Methods

### 2.1. Cr Thin Films Deposition

Copper disks (Electrolytic Tough Pitch purity grade) 20 mm diameter and 4 mm thick were cut from a bar and used as substrates after polishing up to a mirror finish with emery paper and alumina paste suspension (down to 1 μm). Before deposition, they were degreased in isopropyl alcohol using an ultrasonic bath (5 min), pickled in acidic solution ($H_2SO_4$ 10% *v/v*, 3 min), rinsed with bidistilled water and acetone (ACS purity grade), and then dried under nitrogen flow. The absence of embedded abrasive particles was assessed by using an optical microscope.

Electrodeposited chromium layers (ED-Cr) were obtained from a commercial bath (NOVOLIT PROCESS, La Tecnogalvano s.r.l., Novara, Italy) and placed directly into an industrial galvanic line, applying voltages between 3.0 and 3.5 V (DC, potentiostatic condition) until a chrome thicknesses of about 0.8 μm was obtained. Nominal thicknesses were calculated from bath specification and checked, after deposition, by weight gain.

The MS process was realized on a lab-scale Korvus HEX deposition system (Korvus Technology. Ltd., Newington, UK) equipped with DC magnetron sputtering sources, using a 50 mm diameter chromium target (99.95% purity grade). Before deposition, the chamber was evacuated to a pressure lower than $6.3 \ 10^{-3}$ Pa. No electrical polarization of the substrate-holder was applied during the deposition but the substrate was maintained at 300 °C to achieve a smoother surface and better adhesion. The plasma current was set to 150 mA. During deposition (140 min), the flow rate of Argon was 30 standard cubic centimeters per minute (SCCM) and the sample stage was kept rotating to improve the thickness uniformity.

### 2.2. Characterization Methods

The thickness of the coatings was determined through X-ray fluorescence (XRF) measurements by using a Bowman B Series XRF spectrometer (Bowman, Schaumburg, IL, USA) employing a semi-fundamental parameter approach (1-point empirical correction) and an acquisition time of 60 s, 50 kV tube voltage, 0.8 mA tube current, and a collimator of 0.6 mm in diameter.

Quantitative color comparisons were obtained using a 380–780 nm wavelength range portable spectrophotometer CM-700d (Konica Minolta, Tokyo, Japan) equipped with an integrating sphere controlled by SpectraMagicNX software (Konica Minolta, Tokyo, Japan). Color coordinates (L*, a*, b*) were obtained from diffuse reflectance spectra according to the CIE's recommendations [36] using a D65 illuminant, an observer at 2°, and maximum aperture with an angle of incidence θ = 8°. The specular component was included (SCI). The sampled area was 95 $mm^2$ corresponding to circular masks having a diameter of 11 mm (MAV11). Each result was averaged using five independent measurements on different samples spots.

Morphological investigations were carried out by using a Nikon eclipse LV150 optical microscope (Nikon Metrology Europe NV, Leuven, Belgium) and an electron microscope (Hitachi SU3800 SEM, Hitachi High-Tech, Tokyo, Japan). Near surface chemical composition was checked by means of X-ray Photoelectron Spectroscopy (XPS). The experiments were carried out in a vacuum system equipped with a VSW HAC 500 hemispherical

electron-energy analyzer using a non-monochromatic Mg K$\alpha$ X-ray source operating at 120 W power (12 kV $\times$ 10 mA). The peaks were fitted using CasaXPS software employing Gauss–Lorentz curves after subtraction of a Shirley-type background.

X-ray diffraction (XRD) was employed to check the mineralogical nature of the deposits. The spectra were collected at CRIST Centre, University of Florence (Florence, Italy) using a XRD D8 Advance (Bruker Corporation, Billerica, MA, USA) powder diffractometer employing Cu K$\alpha$ (0.154187 nm) radiation in the 2$\theta$ range 30–100$°$, applying a step size of 0.03$°$.

Electrochemical characterization was performed using an Autolab PGSTAT204 potentiostat (Metrohm AG, Herisau, Switzerland) equipped with a FRA32M Frequency Response Analyzer to conduct electrochemical spectroscopy measurements (EIS), controlled by NOVA software and a K0235 flat cell kit (Ametek, Berwyn, PA, USA). A classical three-electrode set-up was employed using a platinum mesh as counter electrodes and an SCE reference electrode separated from the solution by an ion-conducting glass frit. All the experiments were conducted at room temperature (25 $\pm$ 1 $°$C) in a 0.62 M NaCl (3.5 wt.%) aqueous solution and the exposed working electrode surface was 1.0 cm$^2$.

Open circuit potentials (OCPs) were obtained monitoring the potential as a function of time up to one hour. After this period, EIS measurements were carried out at the open circuit potential (E$_{ocp}$), in a 100 kHz to 10 mHz frequency range, with 10 mV of voltage perturbation amplitude. Then, potentiodynamic curves were acquired in a −0.1 to 1.0 Volt range with respect to the E$_{ocp}$ at the scan rate of 0.167 mV/s. Triplicate measurements checked the reproducibility of the presented data, and typical results were reported.

Finally, free corrosion tests were performed on five samples for every type of coating in a DCT600 P salt spray cabinet (ACS Angelantoni, Massa Martana, Italy), spraying a 50 $\pm$ 5 g/dm$^3$ NaCl solution at 35 $\pm$ 2 $°$C (UNI ISO 9227—Corrosion tests in artificial atmospheres. Salt spray tests) for up to 144 h. The corrosion features were then evaluated via visual and optical microscope investigations.

## 3. Results

### 3.1. XRF Thickness Measurement

The homogeneity of a coating's thickness was determined via XRF investigation on a grid constituting 13 points of measurement (Figure 1). As expected, the ED samples presented typical tiny edge effects (Figure 1), while these features were almost absent in the MS deposits. On the other hand, these measurements also show that the central portion of the two series of samples, on which the electrochemical experiments were performed, are very close in terms of thickness, thus allowing for the direct comparison of these results.

### 3.2. Color Measurement

The images displayed in Figure 1 are representative of the different chromium coatings. It is evident that there are some aesthetical differences among them. Both the coatings appeared to be smooth with a metallic lustrous aspect; however, the MS-Cr samples were darker than the ED ones. Regarding this characteristic, accurate color measurements were performed and expressed in terms of L*, a*, and b* parameters (L* =light vs. dark, a* = red vs. green, and b* = yellow vs. blue). In the CIELAB color space (see Figure 2), the a and b coordinates define the hue (color) and the chroma (vividness/dullness), while the degree of lightness is represented by the L* values. Table 1 summarizes the results of this investigation, evidencing that the MS-Cr coatings were darker and more yellowish than the ED ones. Color differences between differently deposited chromium coatings have been already observed and, in many cases, these differences have been attributed to metallic impurities [14] or to the presence of amorphous Cr-phase, graphite, and Cr$_{23}$C$_6$ nanoparticles [37] deriving from organic bath additives. In this case, however, different reasons need to be addressed, since XRD, XPS, and EDS measurements (see the following paragraphs) ruled out the presence of carbides and surface or bulk contaminants.

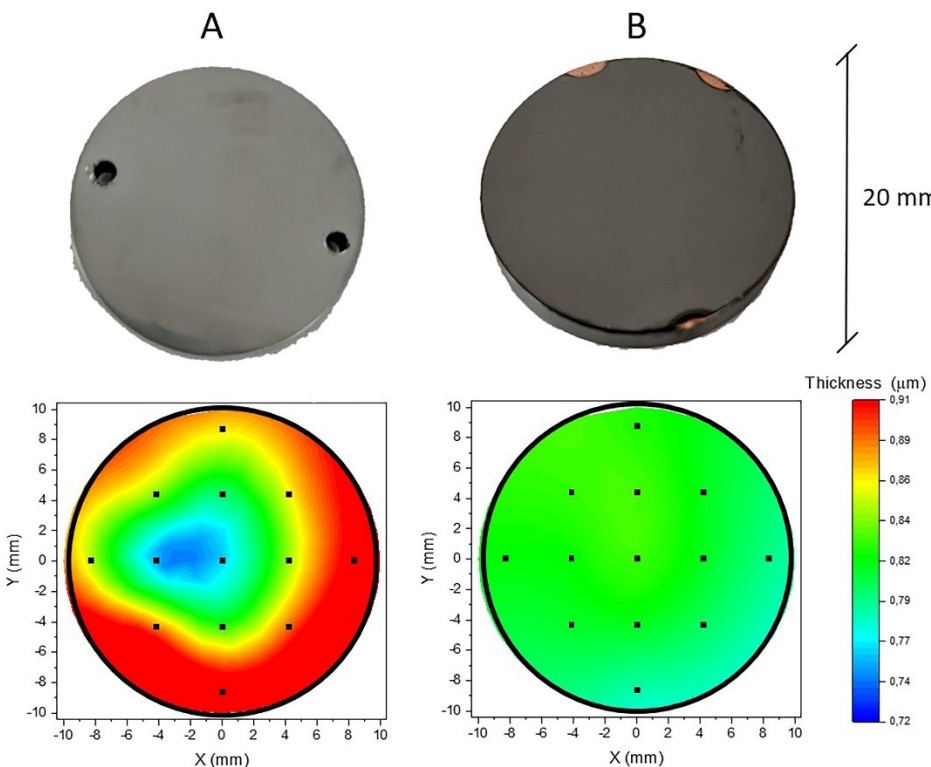

**Figure 1.** Optical images (**above**) and respective thickness contour plots (**below**) of electrodepos-ited (**A**) and magnetron-sputtered (**B**) samples. The black dots in the contour plots depict the grid of points checked by XRF.

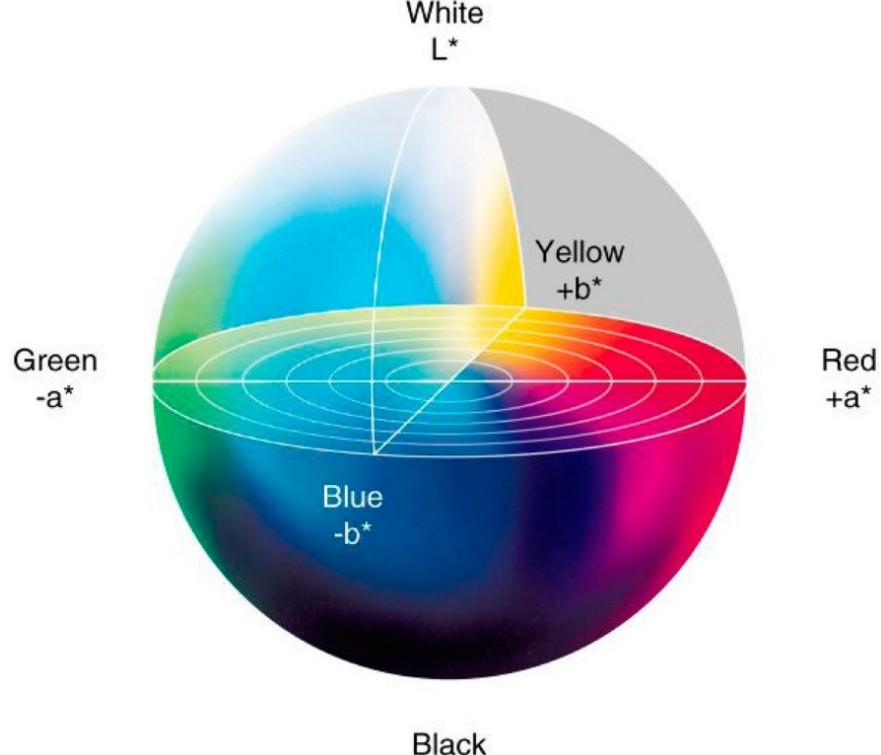

**Figure 2.** CIELAB color space.

**Table 1.** Color parameters characteristic of ED and MS chromium coatings.

| Samples | L* (D65) | a* (D65) | b* (D65) |
|---|---|---|---|
| ED-Cr | 81.74 | −1.01 | −0.96 |
| MS-Cr | 48.95 | 0.90 | 8.34 |

### 3.3. Morphological and Near Surface Chemical Investigation

An insight into the surface morphology of the samples was achieved by using SEM, which revealed significant differences between the ED and MS coatings, as evidenced by the images in Figure 3. Specifically, even they were if chemically identical (checked by EDS, not shown here), the ED-Cr coatings featured globular structures with grains in the range of 400–600 nm. The coatings also presented some defects such as holes and macro cracks. On the other hand, the MS-Cr samples did not show evidence of macro defects and the surface of the coatings exhibited a finer structure not larger than 100 nm, typical of columnar sputtered Cr coatings [37].

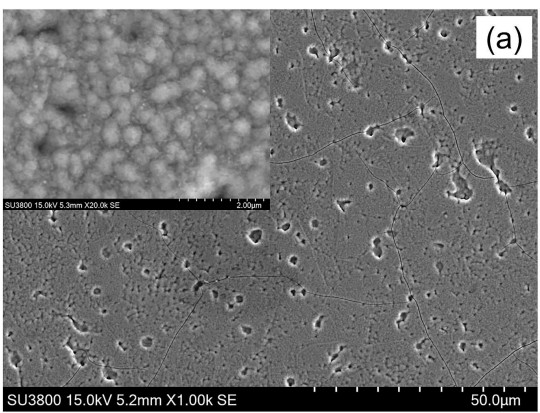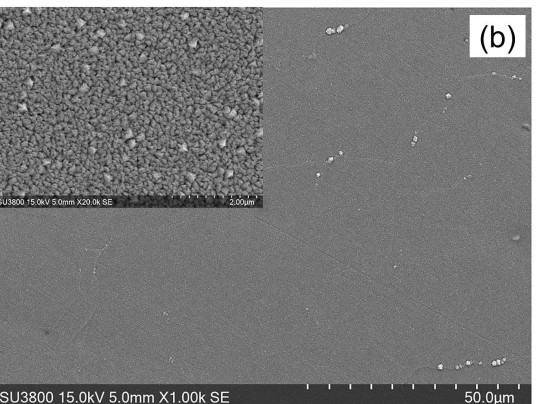

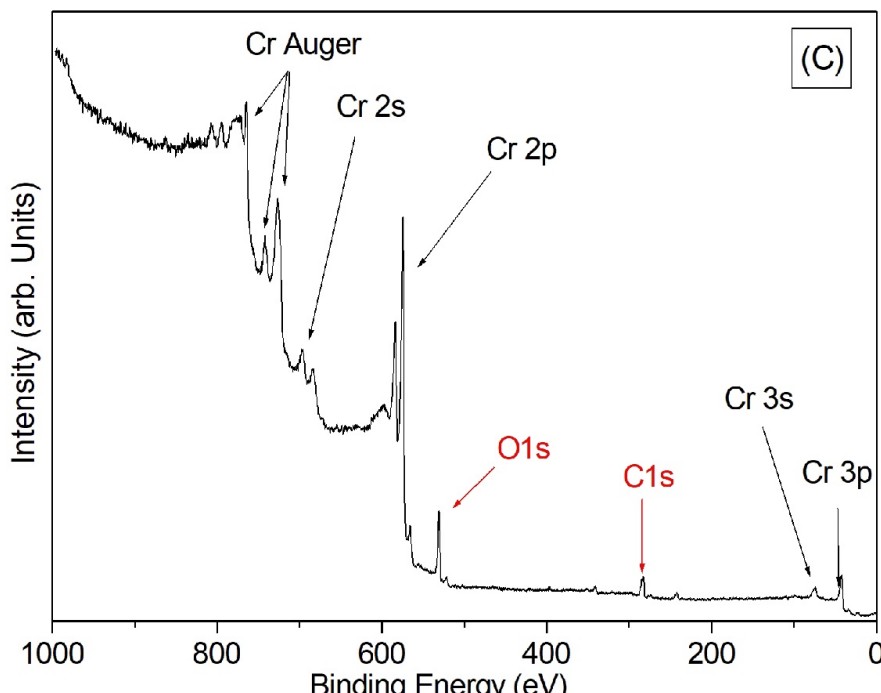

**Figure 3.** SEM images of the surface topography of ED-Cr, (**a**) and MS-Cr (**b**) coating surfaces. XPS survey spectra of MS-Cr coatings evidencing the presence of peaks only attributable to metallic chromium and atmospheric contaminants (carbon and oxygen) (**c**).

Chemically speaking, the surface of these samples was constituted by nearly pure chromium oxides. XPS spectra collected on the MS-Cr sample (Figure 3c) show the absence of all extraneous elements but carbon, which is a well-known ubiquitous atmospheric contamination. Therefore, it is reasonable to attribute the darker appearance of these samples to the finer surface nanostructure of the MS-Cr coatings [15]. This conclusion is supported by several works in which the visual aspect of MS-Cr coatings changed as a function of different microstructures obtained by tuning operating parameters such as plasma power and pressure [23] or by polarizing the sample [38].

### 3.4. X-ray Investigation

Figure 4 illustrates the XRD patterns of the ED-Cr and MS-Cr samples collected in Bragg–Brentano mode. The peaks at 44°, clearly detectable for both samples, are attributed to the (110) plane of crystalline chromium. Other characteristic diffraction peaks of crystalline chromium planes (200) and (211) are barely detectable only for the MS-Cr samples, indicating the strong texturizing of both type of deposits. Substrate (copper) diffraction peaks are also indicated. However, there are no peaks attributable to carbides which sometime result from the reaction of organic additives in the electroplating bath and have been proposed as the reason for yellowish color shifts in the appearance of electrodeposited Cr coatings in the case of chromium(III)-based solutions.

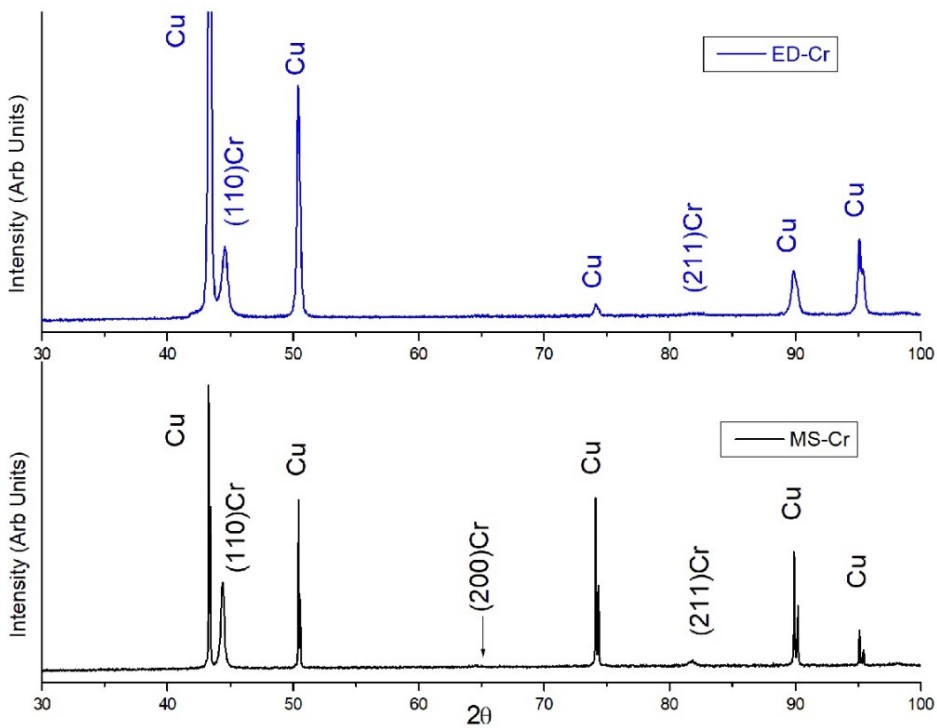

**Figure 4.** X-ray diffraction of the electrodeposited ED-Cr (**above**) and magnetron sputtered (MS-Cr (**below**) chromium coatings. Substrate (copper) diffraction peaks (copper) have also been indicated.

### 3.5. Electrochemical Characterization

#### 3.5.1. OCP

Open circuit potential (OCP) curves were recorded to gain insight into the relative activity of the two chromium coatings. Figure 5 depicts the potential values as a function of time in aerated 3.5% NaCl solution for the two samples compared with the bare substrate (copper). The curves demonstrate the superior behavior of the MS samples with respect to the ED ones, which behaved in a similar way to the substrate (−0.29 V/SCE for ED-Cr and −0.27 V/SCE copper). That is due to the presence of small cracks and pinholes (Figure 3a) in the ED samples which allow the test solution to make contact with the

substrate, promoting an active behavior. Vice versa, the absence of these macrodefects allowed the MS-Cr coatings to exert a superior level of corrosion protection, reaching a less negative plateau potential (−0.01 V/SCE).

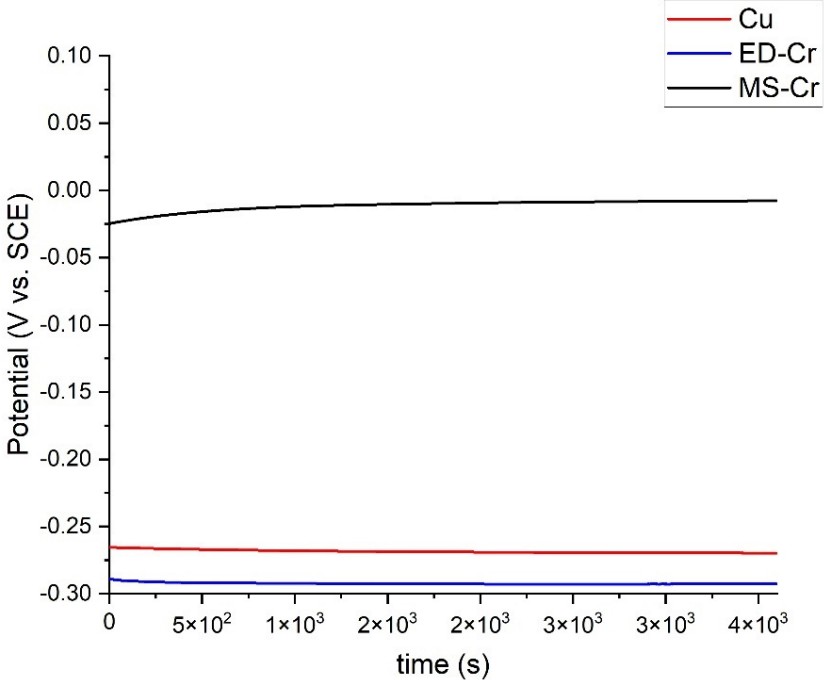

**Figure 5.** Open circuit potential curves (OCPs) of ED-Cr and MS-Cr coatings compared with the bare Cu in aerated 3.5% NaCl solution as a function of time.

3.5.2. Potentiodynamic Polarization

Potentiodynamic polarization (PD) curves were recorded to shed some light on the kinetics of the corrosion processes taking place at the surface. The corresponding curves are displayed in Figure 6 and are compared to pure copper. The ED-Cr samples closely resemble the behavior of copper, the corrosion mechanism of which, in aerated chloride bearing solutions, is a well-known process [39,40]. Specifically, the process starts with the dissolution of metal through the oxidation of Cu (0) to Cu (I). Next, a labile insoluble CuCl layer absorbs at the surface, providing modest corrosion protection, and, finally, Cu (I) oxidizes to Cu (II). On the ground of the similarities in the shapes of the curves collected for Cr-ED samples, it is reasonable that the above-mentioned mechanism also dominates the redox processes taking place at the surface of Cr-ED samples; this is consistent with the presence of macro cracks, as evidenced in Figure 3, which allow the testing solution to make contact with the substrate. However, it is also worth noting that for the ED-Cr samples, the $i_{corr}$ is lower than the pure copper (Figure 6 and Table 2), indicating that the corrosion phenomena are the same overall, despite the extent of the result being reduced for the Cr coated samples. This is attributable to the role played by the Cr-coating in terms of corrosion protection.

The curves collected on the MS coating are characterized by a more positive $E_{corr}$ (Table 2), indicating the superior behavior of this system. On the other hand, the corrosion current, though lower than the pure copper, is higher than in the Cr-ED samples (Table 2). This is probably due to the much finer surface structure of these coatings, which translates onto a larger real surface and, therefore, a higher apparent current density.

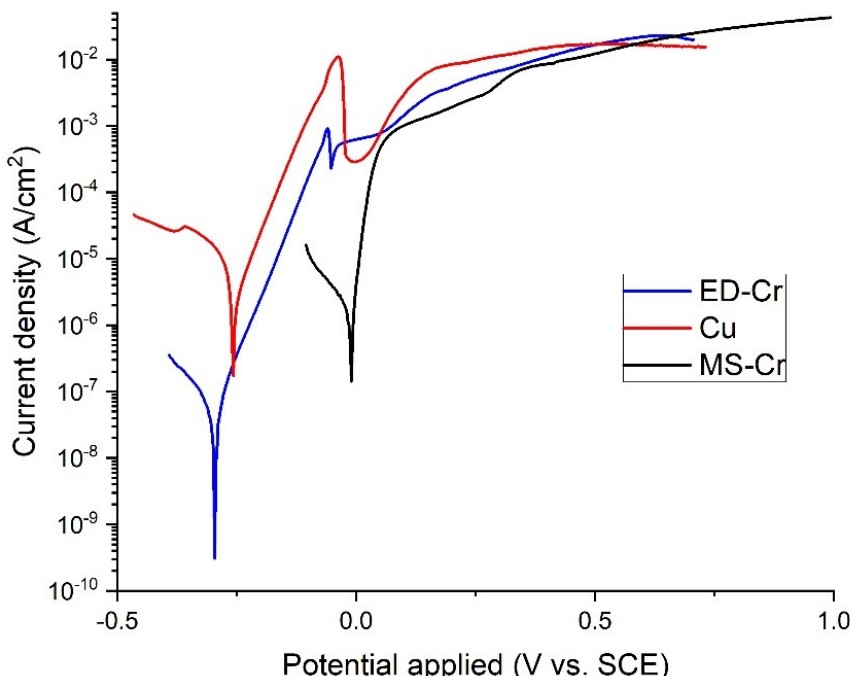

**Figure 6.** Potentiodynamic curves of ED-Cr and MS-Cr chromium coatings compared with the bare copper substrate in aerated 3.5% NaCl solution.

**Table 2.** Electrochemical corrosion data.

| Samples | OCP | $i_{corr}$ (A/cm$^2$) | $\mid b_a \mid$ (V/dec) | $\mid b_c \mid$ (V/dec) | Polarization Resistance $R_{ct}$ (kΩ) |
|---|---|---|---|---|---|
| Cu | $-0.26 \pm 0.01$ | $2.6 \times 10^{-6}$ | 0.06 | 0.07 | 13.5 |
| ED-Cr | $-0.29 \pm 0.01$ | $1.6 \times 10^{-7}$ | 0.06 | 0.15 | 130.7 |
| MS-Cr | $-0.02 \pm 0.01$ | $9.6 \times 10^{-7}$ | 0.02 | 0.06 | 4.8 |

3.5.3. Electrochemical Impedance Spectroscopy (EIS)

Further insight into the interactions between the surface and the solution were gained through investigations by means of EIS. The AC impedance responses of the prepared samples are depicted as Nyquist and Bode view plots in Figure 6 and compared to those of the bare copper substrates. As shown, the Cr-coated samples are characterized by single capacitive loops, with different diameters, while the bare copper substrate present a more complex AC response which requires two constant phase elements [41]. The equivalent circuits used to fit EIS data are depicted in Figure 7 In these circuits, $R_s$ is the solution resistance, $R_{ct}$ is the charge transfer resistance, and $CPE_{dl}$ is the constant phase element, generally attributed to the double layers formed at the electrode/electrolyte interface. Table 2 summarizes the data fitting of the experimental curves depicted in Figures 6 and 8. As shown, the higher values of charge transfer resistance ($R_{ct}$) are related to the Cr-coated samples, which exhibited less corrosion current densities in the potentiodynamic polarization curve (Figure 6), resulting in a good consistency between these measurements.

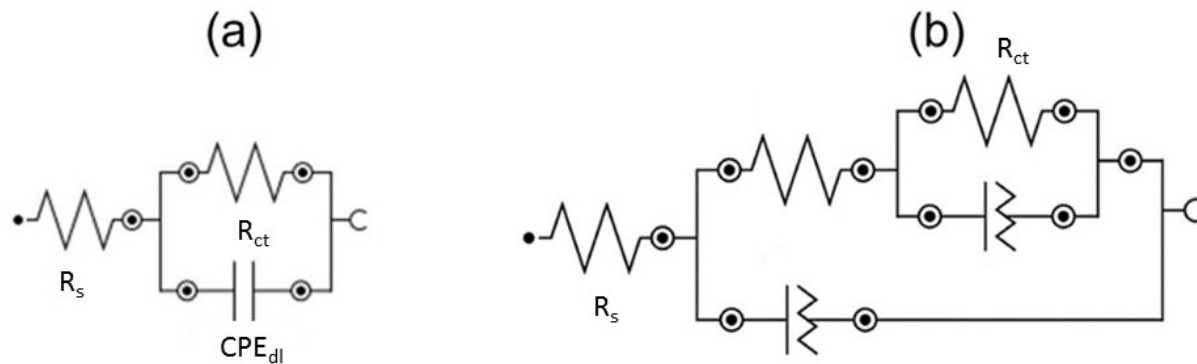

**Figure 7.** Schemes of the equivalent circuits used for modelling the EIS data in Figure 5. Scheme (**a**) was used for Cr-coated and (**b**) for copper substrate.

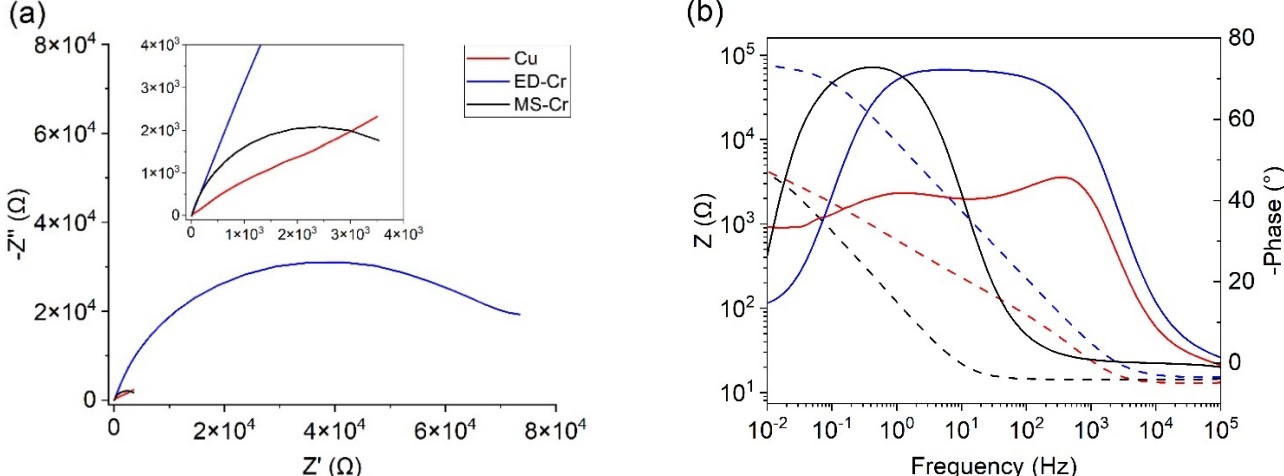

**Figure 8.** EIS data represented as Nyquist plot (**a**) and Bode phase plots (**b**) obtained in aerated 3.5% NaCl solution at room temperature. Samples as in the legend. Inset in the Figure 8a depicts a zoomed view of Nyquist plot.

### 3.6. Free Corrosion Tests

Salt Spray Test

Five samples for each type of coating were prepared, as described in the experimental section, and placed in the salt spray chamber for the free corrosion test (neutral salt spray or NSS test, as described in UNI ISO 9227) for up to 144 h. During this time, the corrosion behavior of the different coatings was monitored by visual investigation. A series of photographs representing the same samples at different exposure times are depicted in Table 3. The photographs evidence that, in this environment, the MS-coated samples present superior corrosion resistance compared to the ED-Cr ones. This is proved by the reduced color variation due to the smaller amounts of corrosion products developed on the surface. This macroscopic result was also confirmed by optical microscope investigation.

The micrographs displayed in Figure 9 show the samples' surfaces after 144 h of exposure to the NSS test. These images highlight the formation of larger amounts of dark and dull corrosion products (mainly constituted by copper oxides) on the surface of the ED-Cr compared to the MS-Cr ones.

**Table 3.** Images of the samples exposed to saline environment as function of time. An ED-Cr sample (above) and a MS-Cr ones (below).

| Time (h) | 0 | 24 | 48 | 72 | 144 |
|---|---|---|---|---|---|
| ED-Cr | | | | | |
| MS-Cr | | | | | |

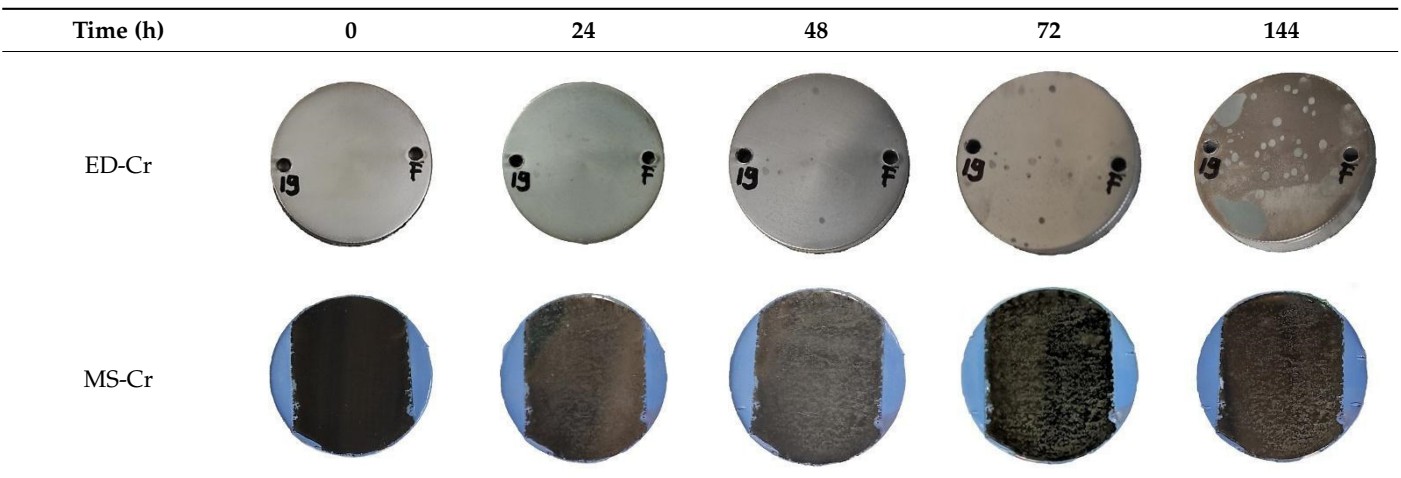

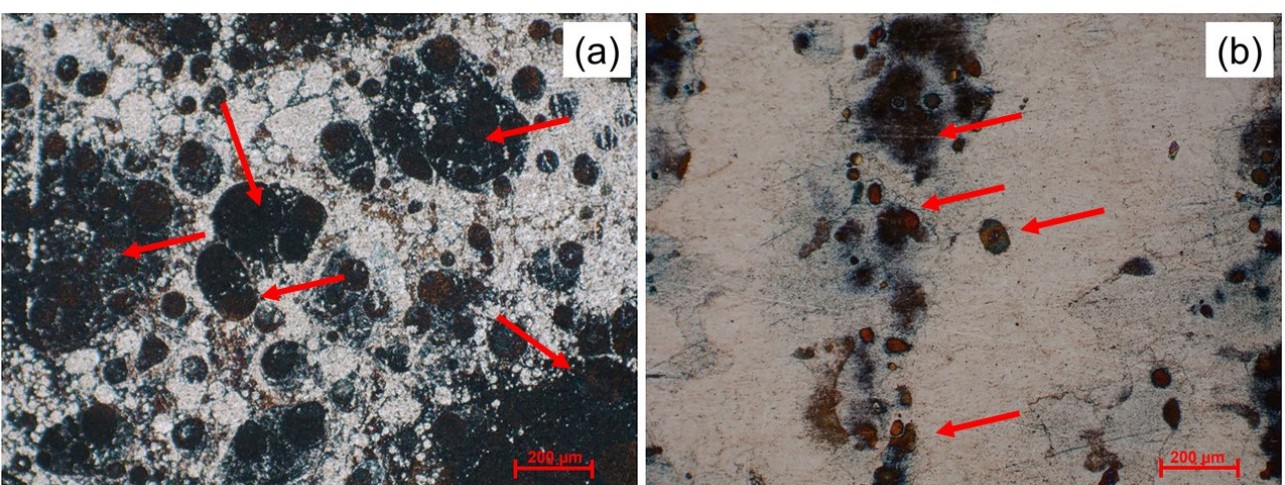

**Figure 9.** Optical micrographs of the ED-Cr (**a**) and MS-Cr (**b**) coatings after free corrosion test (NSS 144 h). The red arrows indicate the locations of corrosion products. Scale bar 200 μm.

## 4. Conclusions

Pure chromium layers 800 nm thick were deposited, via the magnetron sputtering (MS) technique, onto copper substrates and compared with analogous ones, obtained via traditional electrodeposition (ED) from a commercial Cr(VI) plating bath. Overall, even though the MS-Cr coatings were characterized by a darker color, they presented a more homogeneous thickness distribution and similar, if not better, resistance to corrosion in 3.5% NaCl solution, as proved by both electrochemical and free corrosion tests. Indeed, the MS-Cr coatings displayed superior corrosion potential and developed smaller amounts of corrosion products during free corrosion tests.

By means of this comparative investigation in mild condition Cl-bearing aqueous environments, we are able to support the suitability of the chromium magnetron sputtering technique for use as a viable industrial alternative to the traditional technique of hexavalent chromium electroplating for both aesthetic and corrosion protection purposes.

**Author Contributions:** S.M.M. and S.C. conceived and designed the experiments, analyzed the data and wrote the paper; L.D., F.P. and W.G. obtained and characterized the deposits; E.G. and M.I. supervised the research work; N.C. performed the XPS characterization. All authors have read and agreed to the published version of the manuscript.

**Funding:** This research was partially funded by Regione Toscana within the POR FESR TOSCANA 2014–2020 "MetalInk" project (CUP CIPE D64E20003990009).

**Institutional Review Board Statement:** Not applicable.

**Informed Consent Statement:** Not applicable.

**Data Availability Statement:** Not applicable.

**Acknowledgments:** The authors acknowledge Rubinetterie Treemme srl for making their facility available for the realization of the ED samples.

**Conflicts of Interest:** The authors declare no conflict of interest.

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
