# Peer review of "A Comparative Research on Corrosion Behavior of Electroplated and Magnetron Sputtered Chromium Coatings"

_coatings, doi:10.3390/coatings12020257_

Round 1

Reviewer 1 Report

The manuscript presents an interesting study about the corrosion behaviour of chromium coatings deposited using two different methods (MS and ED). Also, the coatings were studied using SEM, XRD, XRF etc. However, the paper needs major revisions before it is processed further, some comments follow:

Abstract

The abstract must be improved. There is no clear purpose of the study in the abstract. Also, please highlight the novelty of the study. The abstract must be reformulated. The abstract must contain information about:

  • Background: Please highlight the purpose of the study;
  • Methods: Describe briefly the main methods used to obtain and characterize the material (OCP, EIS, SEM etc.).
  • Results and conclusions: Indicate the main conclusions or interpretations.

Introduction section

The introduction section must be improved.

Multiple citations have been introduced in bulk form "[9-11]" , "[12-15]" , "[17-19]" and not distributed in the text in accordance with the affirmations that must be supported. Please introduce citation at a specific position to assure a clear correspondence between the affirmations from the introduction section and the previous publication. Moreover, to avoid this type of citing, please cite review type of studies.

Multiple affirmations aren’t supported by the provided references or by the obtained experimental results. Also, please introduce the corresponding citation to support the affirmations from the introduction section. E.g.

The affirmations: " Even if electroplating from Cr(III) aqueous baths would be a straightforward solution, the electroreduction,....”; " By using this non-aqueous media would overcome some of the obstacles....”, " Regarding non-electrochemical methods to deposit chromium coatings, thermal spray could be used for industrial application specially to substitute hard chromium...”, ” purposes. PVD is a process in which the material goes to a vapor phase and then back to a condensed...”, ” A sputtered coating differs fundamentally from...” , ” While many types of sputtering processes are available...”. These affirmations aren’t supported by the provided reference or experimental results. – please introduce corresponding citations.

Materials and methods

Please divide this section into two subsections, one about the deposition methods and process and one about methods used to characterize the coatings obtained.

Results and discussion

Figure 2 c is not clear. Please change it.

Figure 3 is not clear. Please change it.

Figure 5- Please use the same abbreviation for the studied samples. Please change it. Also, in Figure 6.

The results have been presented with very limited discussions. Please improve.

Reference section

Please check carefully the correlation between the cited papers and the position of that reference in the manuscript text body. Some affirmations have no background in published literature.

Author Response

Response to Reviewer 1 Comments:

Abstract: The abstract must be improved. There is no clear purpose of the study in the abstract. Also, please highlight the novelty of the study. The abstract must be reformulated. The abstract must contain information about:

  • Background: Please highlight the purpose of the study;
  • Methods: Describe briefly the main methods used to obtain and characterize the material (OCP, EIS, SEM etc.).
  • Results and conclusions: Indicate the main conclusions or interpretations.

Response: We thanks the reviewer for his/her suggestions. In the light of these considerations we rewrite and amended the abstract. 

Introduction section: The introduction section must be improved.

Multiple citations have been introduced in bulk form "[9-11]" , "[12-15]" , "[17-19]" and not distributed in the text in accordance with the affirmations that must be supported. Please introduce citation at a specific position to assure a clear correspondence between the affirmations from the introduction section and the previous publication. Moreover, to avoid this type of citing, please cite review type of studies.

Multiple affirmations aren’t supported by the provided references or by the obtained experimental results. Also, please introduce the corresponding citation to support the affirmations from the introduction section. E.g.

The affirmations: " Even if electroplating from Cr(III) aqueous baths would be a straightforward solution, the electroreduction,....”; " By using this non-aqueous media would overcome some of the obstacles....”" Regarding non-electrochemical methods to deposit chromium coatings, thermal spray could be used for industrial application specially to substitute hard chromium...”, ” purposes. PVD is a process in which the material goes to a vapor phase and then back to a condensed...”, ” A sputtered coating differs fundamentally from...” , ” While many types of sputtering processes are available...”. These affirmations aren’t supported by the provided reference or experimental results. – please introduce corresponding citations.

Response: The introduction paragraph was rewritten introducing new and more appropriate references. Some imprecisions, as evidenced by the reviewer have been removed.

Materials and methods:Please divide this section into two subsections, one about the deposition methods and process and one about methods used to characterize the coatings obtained.

Response: Done. 

Results and discussion

Figure 2 c is not clear. Please change it.  Response: Substituted.

Figure 3 is not clear. Please change it. Response: Substituted.

Figure 5- Please use the same abbreviation for the studied samples. Please change it. Also, in Figure 6. Response: Done.

The results have been presented with very limited discussions. Please improve. Response: discussion about the results has been extended. We hope it could be considered acceptable.

Reference section.Please check carefully the correlation between the cited papers and the position of that reference in the manuscript text body. Some affirmations have no background in published literature.

Response: The referee is right. Reference list has been accurately checked changing some of the cited papers with more appropriate ones. The number of literature work has also been increased to support a larger number of tecnical aspects. 

Reviewer 2 Report

see attached file

Author Response

Response to Reviewer 2

Response: We wish to thanks the reviewer for the positive opinion about the soundness of our investigation and for the detailed description of the parts to be amended. We hope to have fulfilled these suggestions removing the imprecision noted and improving the quality of the paper.

  • Lines 20, 157, 217, 278, 283: replace »respect« with »with respect« Response: Changed.
  • Line 21: replace »appears« with »appear«Response: Done.
  • Lien 23: replace »resistances« with »resistance« Response: Done.
  • Line 32: replace »layers« with »layer« Response: Done.
  • Line 63: replace »specially« with »especially« Response: Done.
  • Line 67: replace »finishing treatment especially« with »finishing treatment, especially« Response: This part of the text has been completely rewritten.
  • Line 71: replace »id« with »is« Response: Done.
  • Line 81: replace »affect on the« with »affect the« Response: Done.
  • Line 84: replace »have« with »has« Response: Done.
  • Line 89: replace »as: color« with »as color« Response: Done.
  • Line 94: replace were realized onto« with »were deposited onto« Response: Done.
  • Line 97: replace »as industrially« with »as an industrially« Response: Done.
  • Line 108: replace »Volts« with V Response: Done.
  • Line 109: replace »micron« with »micrometer« or »µm« Response: Done.
  • Line 117: replace »process« with »processes« Response: This part of the text has been changed.
  • Line 119: explain the abbreviation XRF Response: The acronim has been elucidated in the text.
  • Line 129: replace »Sampled« with »The sampled« Response: Done.
  • Line 142: replace »1.54187 A« with »0.154187 nm« Response: Done.
  • Line 157: replace »reproducibility« with »reproducibilities« Response: Done.
  • Line 160: replace »in artificial« with »in an artificial« Response: This is the title of the standard test procedure adopted to check the resistance of metallic coatings. Therefore no amendment was introduced.
  • Line 161: replace »microscope« with »microscopes« Response: Changed.
  • Line 165: replace »Coatings« with »Coaating's« and »on a 13« with »on 13« Response: Done.
  • Line 188: replace »showed« with »shown« Response: The text was changed.
  • Line 190: replace »defects as holes« with »defects like as holes«Response: Done.
  • Line 193: at Cr peaks add appropriate labels (e.g. Cr 2p); replace »SEM images of the ED-Cr, (a) and MS-Cr (b) coating surfaces« with »SEM images of the surface topography of ED-Cr, (a) and MS-Cr (b) coatings« Response: Done.
  • Line 201: replace »Microstructural investigation« with »X-ray investigation« Response: Done.
  • Line 203: replace »samples, are« with »samples are« Response: Done.
  • Line 210: replace »Substrate (copper) diffraction peaks (copper) have also been indicated« with »Substrate (copper) diffraction peaks are also indicated« Response: Done.
  • Line 214: replace »were« with »was« Response: Done.
  • Line 219: replace »allow to the« with »allow the« Response: Done.
  • Line 227: replace »about« with »on« Response: Done.
  • Line 228: replace »kinetic« with »kinetics« Response: Done.
  • Line 239: replace »(figure 5 and table 2)« with »Figure 5 and Table 2« Response: Done.
  • Line 256: replace »A further« with »Further« Response: Done.
  • Line 264: replace »layer« with »layers« Response: Done.
  • Line 267: replace »(figure 5)« with »(Figure 5)« Response: Done.
  • Line 277: replace »is« with »are« Response: Done.
  • Line 279: the term »corrosion by-products« in not appropriate Response: it has been substituted with "corrosion products"
  • Line 281: replace »samples« with »sample's« Response: Done.
  • Line 282: replace »NSS test« with »the NSS test« and explain the abbreviation NSS Response: Done.
  • Line 291: the formulation »..compared with chemically and dimensionally analogous ones ..« is strange. Use the better one. Response: This sentence has been rephrased.
  • Line 296: replace »displays« with »display« Response: Done.

General remarks:

  • The process of polishing copper substrates should be described in more detail. How the authors avoided the penetration of abrasive particles into the soft copper substrate during grinding and polishing. The particles built in the substrate can significantly influence the corrosion resistance because they cause the formation of growth defects (see surface irregularities in Fig 2) during the deposition of Cr films. Response: Thanks for this suggeston. Indeed, in the text, (paragrap 2.1)  the preparation of the substrate has been described in more detail. No abrasive particles embedded were observed via microscope investigation. Considering the size of the abrasive suspension used (1 μm) optical microscope check was considered  enough to get an assessment about the clerlness of the surface. 
  • What is the reason for the formation of large pits in ED Cr film Response: Even if there are crack-free Cr coatings, the formation of microcracks on ED-Cr layers occur frequently, expecially whrn commercial electroplating baths are employed. That's sometime attributed to the presence of gas bubbles (generally hydrogen) and/or to the development of microcracks which tend to become larger as the thickness of the layer increases.

Reviewer 3 Report

The article compares two types of chromium coatings, the first deposited trough electrodeposition and the second thanks to magnetron sputtering. Thanks to their approximately similar thicknesses, the authors characterize these two coatings microstructure, structure and performances regarding corrosion. However, the text crucially lacks interpretations, critical review and comparison with already existing work. Indeed, the deposition of chromium by magnetron sputtering or by electrodeposition is well covered, even if it is true that most of the published papers do not deal with both deposition processes at the same time, but one by one. The analysis from the various characterizations is minimal and should be improved before publishing, in order to reach the level expected for a scientific publication. This is frustrating since the authors present a nice amount of results and data, that would be of interest to the community. The goal of the paper is to compare the two deposition processes regarding a chromium coating. Thus the reader expects a certain number of elements regarding the influence of deposition processes, which are lacking. English langage is well used, except for some minor mistakes.

Please find precise remarks in the following, according to their approximate position in the text.

p.2, first paragraph:

In fact three approaches can be selected, the additional one being the use of an alternative material (different from chromium). The second approach is not very well described. We understand that no chromium ions should be involved but the list of potential processes should not be restricted to gas-phase techniques. Thermal spraying, which is mentioned further, is relevant. On the other hand, PVD methods are also relevant, even if some of them use a high content of chromium ions, for instance ionized PVD with cathodic arc deposition and high power impulse magnetron sputtering. The phrasing of the second approach should then be adequately reconsidered.

p.2, l.66-67:

You could cite some examples from the luxury industry, where PVD processes are used to decorate and protect parts from watches.

p.2, l. 69:

I would remove “without the need for ionic species” as it confuses the reader regarding ionized PVD techniques.

p.2, l.71-75:

There is a misunderstanding here. During magnetron sputtering, the matter does not indeed go through a melted state. However, it passes from solid to vapor and is then condensed, as during classical evaporation. The only difference is the way that matter is brought into its vapor state, through sputtering instead of thermal evaporation. The last sentence “a sputtered coating differs fundamentally from an evaporated one because the deposited material has not been physically changed during the deposition” needs to be clarified as it does not sound correct in such shape.

p.2, l.83-84:

You can cite another big application of chromium coating that is undergoing industrial development, the protection of nucleaf fuel cladding. An example of some recent review: https://doi.org/10.3390/coatings10090808

p.3, l.111-118:

Please precise if an electrical polarisation (continuous, voltage?) of the substrate-holder was applied during the deposition. The authors use a reference to justify the heating of the substrates at 300°C. However, the reference relates to a TiN coating deposited on a Ti alloy with certainly different surface roughness. It cannot be extrapolated like this without any further justification.

p.4, l165-170:

Please explain why there is a stronger edge effect for the ED sample, compared to the MS one. The central position of the two samples do not show “exactly” the same thicknesses, but very close ones. Please correct the text accordingly. Moreover, regarding figure 1 and the ED sample, could you please rotate the optical image according to the thickness mapping? So the two holes on the photography are not coinciding with a thickness measurement.

p.5, table 1:

The results from table 1 are not discussed in the text, thus making this table not relevant to display.

p.5, l.190-192:

What the authors refer as “clusters” is very likely to be the top of the columnar microstructure, as it is the typical microstructure observed for sputtered chromium coatings.

p.5, figure 2:

The XPS curve appears in very poor quality, which makes it difficult to read. I suppose that it is due to the compression done by the automatic editing system.

p.6, l.195-199:

As a conclusion to this paragraph, you can also mention that the microstructure of the coatings could be tuned to adjust the visual aspect of the samples. Magnetron sputtering allows tuning the microstructure thanks to the sputtering regime (https://doi.org/10.1016/j.surfcoat.2014.02.030) or more classically the electrical polarization of the sample, as it is well described by structure zone diagrams (https://doi.org/10.1016/j.tsf.2009.10.145).

p.6, l.202-207:

Same remark than for XPS results about the quality. Authors could have used grazing incidence XRD analyses to get rid of the contribution of the substrate. Any comment, comparison or explanation regarding this very strongly textured coatings?

p.7, l.234-240:

The authors explain that the defects present in the ED coating allows the aggressive solution to reach the copper substrate. The next sentence says that the ED chromium coating has an active protection role. These two sentences sound contradictory without further explanations.

p.7, l.244-245:

This hypothesis sounds delicate to evaluate. Please back up this assertion with an adequate reference and possibly calculation of exposed surfaces thanks to roughness measurements coupled with SEM observations.

p.8, l.261-265:

Could you represent of figure 6 the curves calculated with fitted values? It would help to judge the precision and relevancy of the fitting.

p.9, l.282-283:

Please add a legend to the figure 8 to highlight the presence of such corrosion by-products. What is the nature of these by-products? How do you link the performance of the two Cr coatings during salt spray test and during electrochemical measurements?

p.9, l.293-295:

The conclusion is not correct, as the text explained that electrochemical measurements did not show a better performance of MS chromium coatings compared to ED chromium coatings, in terms of charge transfer resistance and corrosion current density.

Author Response

Response to Reviewer 3

Response: First of all we wish to thanks the reviewer for the critical suggestions and meaningfull comments provided about the manuscript. On the base of these considerations we modify the paper removing the imprecisions noted, and  improving the critical comparison of the two type of Cr-coatings, especially regarding the colour differences. We also increased the number of cited literature works as suggested by the referee.

p.2, first paragraph: In fact three approaches can be selected, the additional one being the use of an alternative material (different from chromium). The second approach is not very well described. We understand that no chromium ions should be involved but the list of potential processes should not be restricted to gas-phase techniques. Thermal spraying, which is mentioned further, is relevant. On the other hand, PVD methods are also relevant, even if some of them use a high content of chromium ions, for instance ionized PVD with cathodic arc deposition and high power impulse magnetron sputtering. The phrasing of the second approach should then be adequately reconsidered.

Response: The reviewer is absolutely right. This part of the text is confusing. As a consequence the paragraph has been rewritten to make it more readable.

p.2, l.66-67:

You could cite some examples from the luxury industry, where PVD processes are used to decorate and protect parts from watches.

Response: Ok, thanks for the suggestion. We added reference n°25 and n°26 relative to PVD coatings (not crhoumium) used for easthetical purposes. 

p.2, l. 69:

I would remove “without the need for ionic species” as it confuses the reader regarding ionized PVD techniques.

Response: The text have been rewritten and this confusing sentence removed. 

p.2, l.71-75:

There is a misunderstanding here. During magnetron sputtering, the matter does not indeed go through a melted state. However, it passes from solid to vapor and is then condensed, as during classical evaporation. The only difference is the way that matter is brought into its vapor state, through sputtering instead of thermal evaporation. The last sentence “a sputtered coating differs fundamentally from an evaporated one because the deposited material has not been physically changed during the deposition” needs to be clarified as it does not sound correct in such shape.

Response: The referee is right. The text have been rewritten (from row 80 to row 88) in order to amend these imprecisions. 

p.2, l.83-84:

You can cite another big application of chromium coating that is undergoing industrial development, the protection of nucleaf fuel cladding. An example of some recent review: https://doi.org/10.3390/coatings10090808

Response: Very good suggestion. The application has been introduced and the review paper suggested  added to the reference list.   

p.3, l.111-118:

Please precise if an electrical polarisation (continuous, voltage?) of the substrate-holder was applied during the deposition. The authors use a reference to justify the heating of the substrates at 300°C. However, the reference relates to a TiN coating deposited on a Ti alloy with certainly different surface roughness. It cannot be extrapolated like this without any further justification.

Response: Correct. No electrical polarization was applied. The text (end of paragrap 2.1) was updated and consequently amended. The abovementioned reference was removed.

p.4, l165-170:

Please explain why there is a stronger edge effect for the ED sample, compared to the MS one. The central position of the two samples do not show “exactly” the same thicknesses, but very close ones. Please correct the text accordingly. Moreover, regarding figure 1 and the ED sample, could you please rotate the optical image according to the thickness mapping? So the two holes on the photography are not coinciding with a thickness measurement.

Response: Figure 1 was amended. Edge effects in electroplating process are due to inhomogeneous current density and  odd potential distributions as consequence of Gauss law. Both the electric field and the current density, in presence of sharp tips or edges, are enhanced. Therefore,  thickening of the deposit is regularly observed at the periphery of the electrode  [Newman J. Resistance for flow of current to a disk. J Electrochem Soc. 1966;113(5):501–502]. Such features can be reduced by optimizing the electrodes geometry and the anode/cathode ratio but these can not be completely suppressed, expecially in commercial plating facility. 

p.5, table 1:

The results from table 1 are not discussed in the text, thus making this table not relevant to display.

Response: Correct. The discussion about the observed colour differences has been expanded adding a figure representative of CIELAB colour space (Figure 1) and a more accurate description of the  results obtained in our samples. Now the data displayed in Table 1 are meaningfull. 

p.5, l.190-192:

What the authors refer as “clusters” is very likely to be the top of the columnar microstructure, as it is the typical microstructure observed for sputtered chromium coatings.

Response: Correct. The text (paragraph 3.2, from row 221 to row 223) has been consequently amended.

p.5, figure 2:

The XPS curve appears in very poor quality, which makes it difficult to read. I suppose that it is due to the compression done by the automatic editing system.

Response: An higher resolution figure has been inserted. (Figure 3c)

p.6, l.195-199:

As a conclusion to this paragraph, you can also mention that the microstructure of the coatings could be tuned to adjust the visual aspect of the samples. Magnetron sputtering allows tuning the microstructure thanks to the sputtering regime (https://doi.org/10.1016/j.surfcoat.2014.02.030) or more classically the electrical polarization of the sample, as it is well described by structure zone diagrams (https://doi.org/10.1016/j.tsf.2009.10.145).

Response: We thanks the referee for this usefull suggestion. The manuscript has been amended consequently and the papers recommended have been added to the reference list. 

p.6, l.202-207:

Same remark than for XPS results about the quality. Authors could have used grazing incidence XRD analyses to get rid of the contribution of the substrate. Any comment, comparison or explanation regarding this very strongly textured coatings?

Response: XRD measurements were carried out to check the cristalline structure of the two Cr-coatings and the absence of anomalous cristalline phases (es carbides). The strong texture observed on both the coatings, is not surprising. Several papers already deals with texture observed on PVD or electrodeposited chromium coatings (see for example C. Bergenstof N. et al J. appl. electrochem. 28 (1998) 141). Terefore this aspect is not new and it was not investigated further. 

p.7, l.234-240:

The authors explain that the defects present in the ED coating allows the aggressive solution to reach the copper substrate. The next sentence says that the ED chromium coating has an active protection role. These two sentences sound contradictory without further explanations.

Response: Correct. The sentence has been rephrased to better explain the role played by Cr layer. (row 278 - 280).

p.7, l.244-245:

This hypothesis sounds delicate to evaluate. Please back up this assertion with an adequate reference and possibly calculation of exposed surfaces thanks to roughness measurements coupled with SEM observations.

Response: I am sorry but I am not sure to have fully understood this comment. The effect of real surface on electrochemcial results is a well known limit of electrochemical corrosion tests (see for example the classic work of De Levie Elect. Acta 10, 1965, Pages 113-130). Being impossible to compare Cr-coatings characterized by the same surface roughness, we carried out parrallel free corrosion tests in the same aggressive environment (areated 3,5% NaCl solution). In such a way the electrochemcial tests returned an idea about the corrosion tendency and the mechanism of the corrosion processes (noble or active behaviour (OCP), passive /transpassive tendency (polarization curves), interface and double layer stability (EIS)) while NSS test provide the extent of such phenomena. It seems reasonable to attribute the corrosion rate discrepancies observed between these different investigation techniques to a larger (or smaller) contribute of the real sample's surface.

p.8, l.261-265:

Could you represent of figure 6 the curves calculated with fitted values? It would help to judge the precision and relevancy of the fitting.

Response: We thank the referee for this suggestion. In principle it is possible. However, by adding the fitting curves to these graphs would make them much more complex. In order to mantain the figure easily readable we would prefer to mantain these graphs without the fitting lines. Of course, if this aspect is considered mandatory, the figures can be modified accordingly.

 p.9, l.282-283:

Please add a legend to the figure 8 to highlight the presence of such corrosion by-products. What is the nature of these by-products? How do you link the performance of the two Cr coatings during salt spray test and during electrochemical measurements?

Response: Arrows indicating the locations of corrosion products have been inserted in Figure 9. The different performances of the two type of coatings have been attributed to different real coatings surface 

p.9, l.293-295:

The conclusion is not correct, as the text explained that electrochemical measurements did not show a better performance of MS chromium coatings compared to ED chromium coatings, in terms of charge transfer resistance and corrosion current density.

Response: Correct. However, both these parameters are affected by real electrode area which differ from ED-Cr and MS-Cr coatings.

Round 2

Reviewer 1 Report

The author addressed all of my comments. The manuscript can be accepted for publication in the present form. 

Author Response

We thanks the reviewer for his/her positive comments. 

Reviewer 3 Report

I would like to thank the authors for this revised version and all the associated efforts. I recommend publication of this manuscript, which will be of big interest for the community.

Author Response

We wish to thank the referee for his/her positive comment on our manuscript